# NF-κB as an Inducible Regulator of Inflammation in the Central Nervous System

**DOI:** 10.3390/cells13060485

**Published:** 2024-03-11

**Authors:** Sudha Anilkumar, Elizabeth Wright-Jin

**Affiliations:** 1Neonatal Brain Injury Laboratory, Division of Biomedical Research, Nemours Children’s Health, Wilmington, DE 19803, USA; 2Division of Neurology, Department of Pediatrics, Nemours Children’s Health, Wilmington, DE 19803, USA; 3Department of Psychological and Brain Sciences, University of Delaware, Newark, DE 19716, USA; 4Sidney Kimmel Medical College, Thomas Jefferson University, Philadelphia, PA 19107, USA

**Keywords:** NF-κB, inflammation, microglia

## Abstract

The NF-κB (nuclear factor K-light-chain-enhancer of activated B cells) transcription factor family is critical for modulating the immune proinflammatory response throughout the body. During the resting state, inactive NF-κB is sequestered by IκB in the cytoplasm. The proteasomal degradation of IκB activates NF-κB, mediating its translocation into the nucleus to act as a nuclear transcription factor in the upregulation of proinflammatory genes. Stimuli that initiate NF-κB activation are diverse but are canonically attributed to proinflammatory cytokines and chemokines. Downstream effects of NF-κB are cell type-specific and, in the majority of cases, result in the activation of pro-inflammatory cascades. Acting as the primary immune responders of the central nervous system, microglia exhibit upregulation of NF-κB upon activation in response to pathological conditions. Under such circumstances, microglial crosstalk with other cell types in the central nervous system can induce cell death, further exacerbating the disease pathology. In this review, we will emphasize the role of NF-κB in triggering neuroinflammation mediated by microglia.

## 1. NF-κB Signaling Pathway

The nuclear factor kappa B (NF-κB) family are the principal transcription factors involved in the regulation of immune-mediated inflammatory responses. Several hundred NF-κB target genes have been identified, with great diversity in function including cytokine or chemokine modulation, apoptosis, and cell proliferation [1,2]. The NF-κB signaling pathway has been implicated in a number of disorders, encompassing neurodegenerative diseases, autoimmune disorders, cancer, and metabolic disorders [1,2]. NF-κB signaling can be divided into canonical (classical) and noncanonical (alternative) pathways, distinguished by the nature of stimuli and mechanism of activation. The canonical pathway is regarded as the central regulator of the inflammatory response and it has been extensively studied in the context of human disorders [3]. Canonical NF-κB signaling is rapidly mounted in response to acute inflammatory cues and is involved in mediating the innate immune response [1,2]. Conversely, noncanonical signaling responses are typically slow and sustained, characteristic of the response to immune dysregulation [4]. Noncanonical signaling is associated with the adaptive immune response, including lymphoid organogenesis along with B cell survival and differentiation [4,5]. Fundamentally, NF-κB not only plays a diverse role in immune responses but is significant to elucidating the mechanisms underlying several diseases.

The NF-κB family comprises 15 identified homo- or heterodimers originating from five protein subunits: p65 (RelA), c-Rel, RelB, p50/p105 (NF-κB1), and p52/p100 (NF-κB2) [1,6]. All NF-κB subunits are structurally similar and possess a Rel Homology Domain (RHD) [7,8]. The highly conserved RHD is composed of a DNA binding domain at the N-terminal and a site for interaction with inhibitors at the C-terminal [8]. The NF-κB subunits can be broadly categorized into Class I precursors (p105, p100) and Class II Rel proteins (RelA, RelB, and c-Rel), distinguished by their possession of transcription transactivating domains (TADs) [1]. Class I precursors undergo C-terminal cleavage during processing before maturation into active p50 (NF-κB1) and p52 (NF-κB2) proteins [9] (Figure 1).

Under homeostatic conditions, NF-κB is inactivated and sequestered to the cytoplasm either by natural inhibitors of nuclear factor kappa B (IκB) proteins (primarily IκBα, IκBβ, and IκBε) or by inherent structural inactivation [3]. Upon NF-κB activation, two discrete signaling pathways, canonical (classical) and non-canonical (alternative) pathways, can be stimulated in nearly all cell types within the human body (Figure 2). Nearly all NF-κB dimers participate in canonical signaling, with the RelA-p50 heterodimer as the most abundant NF-κB transcription factor [1,2,3]. Within noncanonical signaling, the primary NF-κB dimer is RelB-p52 [4,5]. One key pathway difference between the canonical and noncanonical NF-κB formation lies in the order of dimerization and processing events. In canonical signaling, the p105 precursor is first proteolyzed to form mature NF-κB1 and then bound to p65/RelA [3,10,11]. In noncanonical signaling, the p100 precursor is initially bound to Rel B before undergoing processing to form mature NF-κB2-RelB [4,5] (Figure 2). At this point, these mature NF-κB dimers can undergo translocation to the nucleus to regulate the transcription of target genes.

The canonical NF-κB signaling pathway can be activated by a diverse array of inflammatory stimuli. Typically, these stimuli include bacterial or viral products (such as lipopolysaccharide or viral proteins), inflammatory cytokines (primarily TNFα, IL-1, and IL-6), and physical or chemical stressors [1,4,12]. Within immune cells, which express pattern recognition receptors (PRRs), NF-κB activation occurs in response to pathogen-associated molecular patterns (PAMPs) or damage-associated molecular patterns (DAMPs) released by damaged or dying cells [12,13]. In the canonical NF-κB pathway, upstream signaling for such stimuli converges at the stimulation of transforming growth factor-beta-activated kinase 1 (TAK1) [13]. TAK1 subsequently phosphorylates the IκB kinase (IKK) complex, prevalently considered the master regulator of the NF-κB pathway. The activated IKK complex mediates the phosphorylation of IκBɑ at a serine residue for ubiquitination. The ubiquitination of IκBα targets this protein for degradation by the proteosome, a multi-subunit protease complex localized to the cytoplasm. Without the inhibitor protein, the final NF-κB product can be translocated to the nucleus [10,13]. Once in the nucleus, NF-κB can bind to its specific DNA consensus site (κB site) of various genes participating in inflammation or mediating the immune response. Within inflammatory contexts, the gene products of NF-κB1 activation in a given cell can induce NF-κB signaling in neighboring cells, propagating the response to the harmful stimulus [14]. Beyond immune response, NF-κB1 is implicated in many processes depending on cell type, including differentiation, regulation of apoptosis, and even metabolic responses [11].

Noncanonical NF-κB signaling is typically slower to activate and sustained in its response [14]. The noncanonical pathway is triggered by more specific stimuli, namely members of the tumor necrosis factor (TNF) superfamily (which also induces canonical NF-κB signaling), CD40, and RANKL [14,15]. The mechanism of action for noncanonical NF-κB signaling is also distinct. Cellular responses to the aforementioned stimuli converge with activation of NF-κB-inducing kinase (NIK), which subsequently activates IKKα via phosphorylation [14]. Fundamentally, the pathway is reliant on the processing of p100, which dually functions as a precursor to p52 and an inhibitor of mature NF-κB translocation [14]. Therefore, p100 serves as the inducible regulator of the noncanonical pathway, mimicking the role of IκB in the canonical pathway. Much like in canonical signaling, a member of the IKK protein family (IKKα) mediates p100 phosphorylation and eventual ubiquitination for proteasomal degradation [14]. At this point, the active NF-κB2 molecule (comprising p52 and RelB subunits) can be translocated into the nucleus for the differential regulation of genes involved in inflammation, cell survival, and immune cell development [15].

## 2. NF-κB in the Central Nervous System

NF-κB plays a multifaceted role within the central nervous system, with diverse roles in each cell type that vary depending on physiological conditions [11]. Within neurons, NF-κB activation is essential for information processing and transmission and can produce both neuroprotective and neurodegenerative effects [5,16,17,18]. Effects of NF-κB activation differ based on glial cell type as well. In oligodendrocytes, NF-κB has the potential to promote cell survival [19,20,21]. However, NF-κB activation in astrocytes and microglia typically is associated with more detrimental outcomes [22]. An overview of the functions of NF-κB signaling are outlined in Table 1.

### 2.1. Neurons

Broadly, NF-κB is intricately involved in several disparate aspects of neuron cell function, comprising both neural behavior under normal physiological conditions and neuron survival under disease conditions. NF-κB involvement has been documented in synaptic plasticity, growth factor signaling, and even higher order cognition (including learning and memory) [48]. The function of NF-κB in regulating the function of neurons under physiologic conditions is outside the scope of this review; however, several recent reviews [11,22,48] offer a more detailed exploration of this topic. NF-κB is a key modulator of neuron survival, with a dual capacity to be both neuroprotective and neurodegenerative [5,16,17,18].

Typically, neuronal NF-κB signaling is protective. NF-κB selectively induces anti-apoptotic genes such as TNF receptor associated factors (namely TRAF-1 and TRAF-2), caspase inhibitors, apoptosis regulators in the Bcl-2 family, and superoxide dismutase (SOD) [5,46,49,50,51]. The neuroprotective effect of NF-κB is more explicitly represented in neurons derived from rodent models and subjected to apoptosis-promoting conditions. Within cultured embryonic rat hippocampal neurons, NF-κB activation by TNF-α and ceramide increased neuron survival under oxidative stress (FeSO_4_ and amyloid beta peptide treatment) [5]. These results were further confirmed when the neuroprotective effects were lost after the introduction of decoy DNA to deactivate NF-κB [5]. The anti-apoptotic role of NF-κB is further corroborated in a transgenic mouse model with forebrain neuron-specific expression of a dominant negative mutant of IκB resulting in the inhibition of NF-κB. As examined in organotypic hippocampal slices, there was a subsequent increase in neuron cell death when exposed to neurotoxic insults (namely FeSO_4_ or kainate) [16].

Moreover, constitutive NF-κB activation has been shown to be necessary for neuron survival in select cases. Within primary cultured cortical neurons, an adenovirus encoding an IκB super-repressor was introduced for the inhibition of neuronal NF-κB; under these circumstances, there was a significant reduction in cell survival. Adenovirus-induced overexpression of p50 (Rel A) in the same model resulted in increased cell survival due to the accumulation of protein protectors against neuronal apoptosis due to etoposide and camptothecin [26]. A similar effect was observed in primary cultured embryonic mouse motor neurons. When interference RNA was used to selectively downregulate IKKα, IKKβ, or RelA proteins necessary for NF-κB activation, motor neuron apoptosis was induced [27]. NF-κB activation is also critical to the survival of developing sensory neurons. Embryos lacking p65 yield significantly fewer sensory neurons in culture [28]. Therefore, NF-κB activity may be intrinsic to the survival of particular neuron populations.

Conversely, some studies describe a neurodegenerative role of NF-κB signaling. A potential mechanism for neuronal cell death via endogenous NF-κB can be attributed to the induction of p53 via TNF-α activated NF-κB [52,53]. Tumor suppressor p53 has been associated with neuronal cell death: p53 knockout mice displayed decreased neuronal apoptosis in response to kainic acid excitation [17]. Furthermore, the TNF-α/NF-κB/p53 axis was identified by transcriptomic analysis as a contributor to cell death in human pluripotent stem cell (hPSC)-derived dopaminergic neurons engrafted into the striatum of mice. When TNF-α/NF-κB signaling was chemically inhibited, there was an enhanced survival of engrafted neurons, experimentally validating the pathway as a mediator of neuronal cell death [18].

The negative effect of NF-κB within neurons has been primarily observed in rodent models of ischemia. Nuclear NF-κB p50 and p65 localization was heightened in hippocampal neurons at the time of cell death in a rat model of global ischemia [25]. These results were further corroborated in human stroke patients, wherein activated NF-κB was detected in neurons at penumbral sites of the sampled brain sections [24]. However, there is conflicting evidence in studies evaluating p50 knockout mice for neuronal degeneration, wherein NF-κB was found to have a positive effect in a model of ischemia and a negative effect in a model of stroke [23,54]. These opposing results could be attributed to potential differences in the subunit composition of expressed neuronal NF-κB. Within cerebellar granule cells, the p65 subunit was implicated in neuronal cell death, whereas c-Rel was found to be essential for cell survival in the case of glutamate-induced excitotoxicity [55]. Therefore, the causal relationship between NF-κB and neuronal cell death needs further investigation.

### 2.2. Glia

NF-κB signaling in glia is primarily associated with inflammation secondary to a pathologic condition, whether as disease or trauma. Within the body, inflammation is the prototypic immune response to any fluctuation from homeostasis in efforts to eradicate abnormal stimuli and promote recovery. The CNS follows the same parameters, wherein microglia, astrocytes, and infiltrating leukocytes (in response to inflammatory conditions) primarily induce inflammation via NF-κB.

#### 2.2.1. Oligodendrocytes

Present research remains divided regarding the effect of NF-κB activation in both healthy and diseased oligodendrocytes. NF-κB demonstrates a protective role in some studies, while other studies suggest a more dispensable role for NF-κB. Activation of NF-κB in oligodendrocyte precursor cells was found to decrease cell apoptosis and promote cell maturation, thereby indirectly contributing to myelination in the CNS [29]. In contrast, in a study involving the CNS-wide deletion of RelA, histological and electron microscopic analysis of the optic nerve showed unimpaired oligodendrocyte densities and normal myelin sheath formation, suggesting that NF-κB is expendable in oligodendrocyte function [56]. Moreover, chronic NF-κB activation in mature oligodendrocytes is found to promote inflammatory conditions in the CNS similar to those in an aging brain [30]. In transgenic mice with constitutively active IKK2, RNA-Seq analysis revealed that the primary oligodendrocytes had gene expression signatures associated with increased post mitotic cellular senescence. These mice had increased white matter degeneration and myelination deficits characteristic of an aging brain [30]. Overall, existing studies demonstrate conflicting effects for NF-κB in healthy oligodendrocytes; additional studies must be conducted to establish the causal relationship of NF-κB in oligodendrocyte survival.

The NF-κB pathway has been more robustly explored in the context of multiple sclerosis (MS), particularly with respect to protection against inflammation, remyelination, and oligodendrocyte survival. In vitro studies in oligodendroglial cell lines revealed that the plasmid-based activation of NF-κB decreased apoptosis, whereas the inhibition of NF-κB increased cytotoxicity under inflammatory conditions caused by TNF-α, IFN-γ, or reactive chemical species [19,20]. These results are corroborated in an experimental autoimmune encephalomyelitis (EAE) mouse model of MS characterized by ectopic expression of IFN-γ in the CNS and an oligodendrocyte-specific expression of a dominant negative mutant of IκB for NF-κB super-repression [21]. The inactivation of NF-κB exacerbated oligodendrocyte death and induced hypomyelination in developing mice or remyelination failure in adult mice [21]. However, one study evaluating the oligodendrocyte-specific deletion of IKK2 for NF-κB inhibition in an EAE mouse model shows comparable numbers of oligodendrocyte progenitor cells and mature oligodendrocytes to the wildtype control, suggesting a more expendable role for NF-κB [31]. A potential cause for the conflicting evidence is that NF-κB activation in oligodendrocytes has also been traced to pancreatic endoplasmic reticulum kinase (PERK) signaling, an IKK-independent pathway in models of EAE and MS [19,57,58]. Therefore, in models of NF-κB inhibition via the deletion of upstream proteins in the canonical pathway, there is still potential for residual NF-κB activation via the PERK pathway. Taken together, the data suggest a protective role for NF-κB activity in oligodendrocytes under acute inflammatory conditions.

#### 2.2.2. Astrocytes

Mirroring the multifaceted functions of astrocytes in the central nervous system, the activation of NF-κB in astrocytes exhibits a similar diversity. NF-κB has been implicated in the astrocyte-dependent clearing of synaptic glutamate, metabolic control, and modulation of astrocyte structural plasticity, as further described in two recent reviews [11,59]. Under normal physiological conditions, NF-κB is involved in astrocytic differentiation from neural progenitor cells (NPCs). Inhibition of the NF-κB pathway in NPCs using suppressors at both the IKK degradation and nuclear translocation stages yielded a significant decrease in viable astrocyte numbers [35]. Additional investigation is necessary to better elucidate the mechanisms for NF-κB-based astrocytic differentiation.

In disease conditions, astrocytic NF-κB signaling is predominantly associated with increased inflammation [32,33,34]. NF-κB activity is upregulated in rodent models of brain and spinal cord injury. Within a model of focal brain injury, increased astrocytic NF-κB expression was identified in early-stage injury as determined by immunohistochemical staining for p50 and p65 subunits [32]. Often, this increase in NF-κB expression is associated with adverse outcomes, particularly in the case of chronic activation within astrocytes. Many studies have investigated the effect of inhibiting NF-κB activation in astrocytes. Within a mouse model of contusive spinal cord injury, selective inactivation of astrocytic NF-κB via transgenic IKKBα inhibition resulted in reduced lesion volume, white matter preservation, and decreased expression of proinflammatory cytokines [33]. In a similar study of spinal cord injury, decreased NF-κB activity promoted the sparing of spinal circuits involved in locomotion with improved functional outcomes [34]. Generally, astrocytic NF-κB activation is associated with increased inflammation within the brain, often worsening pathologic outcomes.

#### 2.2.3. Microglia

As the resident macrophages of the CNS, microglia are typically regulated by NF-κB in the context of inflammation. While microglia can adopt both pro- and anti-inflammatory phenotypes, NF-κB activation is primarily associated with a pro-inflammatory state. During the next section of this review, we will provide an in-depth exploration of microglial NF-κB activity and subsequent crosstalk with other cell types in several pathologies common to the CNS.

As recent research expounds on microglia from a nonimmune standpoint in the healthy brain, the NF-κB pathway has been highlighted in microglial differentiation and homeostasis. In zebrafish, programmed cell death protein 11 (PDCD11) knockout yielded decreased macrophage differentiation into microglia that was reversable by the co-over-expression of c-Rel and p105 [46]. Another study revealed the critical role of NF-κB in establishing the homeostatic density of microglia: conditional knockout of IKKβ (NF-κB inhibition) in a microglia-depleted mouse model showed impaired microglial repopulation and an inability to regain homeostatic density [47]. At present, microglial NF-κB activation in brain development and microglial differentiation represent an under-researched niche, with potential applications in regulating microglial number within the CNS to remediate pathological conditions.

## 3. Role of Microglial NF-κB in CNS Disease

### 3.1. Alzheimer’s Disease

Alzheimer’s disease (AD) is primarily defined by a gradual loss of memory, leading to a spectrum of cognitive and behavioral deficits. Primary contributors to AD pathogenesis are amyloid beta aggregates and tau tangles, neurofibrillary tangles composed of insoluble tau fibrils [33]. This accumulation leads to widespread neuroinflammation and subsequent neurodegeneration, largely proposed to be mediated by microglia [34].

Pathway analysis has identified NF-κB signaling as one of the most perturbed pathways in late-onset AD [60]. NF-κB has been implicated in both amyloid beta plaques and tau fibrils, each mediated in part by microglial activation. NF-κB and beta amyloid plaques have long been proposed to have a positive feedback relationship: the presence of plaques stimulates NF-κB in neurons, which causes the further production of plaques in a degenerative loop [36,37]. Amyloid beta plaques stimulate microglial NF-κB pathways, potentially through extracellular signal-regulated kinase (ERK) and mitogen-activated protein kinase (MAPK) signaling [61,62]. This NF-κB activation causes significant increases in proinflammatory gene expression, generating a neurodegenerative environment. Notably, in the early stages of AD, microglia can mediate phagocytosis of these plaque aggregates. The constitutive activation of NF-κB in astrocytes in a transgenic mouse model of AD caused the polarization of microglia towards a state favoring plaque clearance [63]. However, internal NF-κB activation in microglia typically relates to worsened outcomes in the context of AD.

Emerging research has implicated NF-κB activation in microglia as a key driver of tau tangles, as well [38]. Within a comprehensive transcriptional evaluation of primary microglia derived from a mouse model of tauopathy, NF-κB was among the top affected pathways, along with a demonstrated upregulation in proinflammatory cytokines (IL-1B, TNF, and IL-12B) [38,64]. Single-cell RNA sequencing analysis revealed that NF-κB target genes were differentially expressed in disease-associated microglia, with IκB kinase identified as the top upstream regulator [38]. The knockout of microglia-specific NF-κB in a mouse model of rodent tauopathy prevented spatial learning and memory deficits characteristic of AD [38]. Remarkably, within this study, microglial NF-κB activation dually led to significant increases in tau seeding and spreading in tandem with improved tau fibril clearance [38]. Fundamentally, further research into microglial NF-κB activation in AD is needed to fully understand the multifunctional role of NF-κB. With a clearer insight into the molecular mechanisms governing these NF-κB-mediated interactions, there is a potential avenue for NF-κB as a target in neurotherapeutic development.

### 3.2. Amyotrophic Lateral Sclerosis

Amyotrophic lateral sclerosis (ALS) is a neurodegenerative disease marked by progressive motor neuron death. Clinical presentation of the disease includes gradual impairments in voluntary muscle movements, muscular atrophy, and eventual death due to dysphagia and dyspnea [65]. ALS has been associated with mutations in several genes: superoxide dismutase 1 (SOD1), TAR DNA-binding protein 43 (TARDBP), fused in sarcoma (FUS), TANK-binding kinase 1 (TBK1), and chromosome 9 open reading frame 72 (C9orf72) [65]. While motor neurons are the ultimate effectors of ALS, glial cells have been explored as potential mediators for neurodegeneration, with a particular focus on microglial activation in recent years [66,67].

Microglial NF-κB interactions have been documented in several models of ALS. Spinal cord extracts from sporadic ALS human patients were used to identify TARDBP colocalization with the p65 NF-κB subunit in microglial cells, suggesting that the canonical NF-κB pathway may contribute to TARDBP deregulation characteristic of ALS [39]. Furthermore, within a mouse model of ALS characterized by mutant SOD1, the inhibition of microglial NF-κB rescued motor neurons in a subsequent ALS co-culture model in vitro and significantly increased mouse survival rates in vivo [40]. Selective NF-κB inhibition in this model decreased pro-inflammatory microglia markers (CD68, CD86, and iNOS), suggesting a direct role of NF-κB in ALS pathogenesis [40]. Some studies have suggested that astrocytic NF-κB activation may induce pro-inflammatory microglial proliferation in models of late-stage ALS [41,64]. Remarkably, within the SOD1 mouse model, astrocytic NF-κB inhibition via both transgenic and viral IκB suppression was not sufficient to improve motor neuron survival or ALS progression, highlighting the role of microglial NF-κB in inducing ALS presentation [40,41]. Overall, NF-κB remains a promising target for the attenuation of microglia-mediated inflammation, which may improve outcomes for ALS patients.

### 3.3. Hypoxic Ischemic Encephalopathy

Perinatal hypoxic ischemic encephalopathy (HIE) is a major cause of neonatal death and long-term disability in term infants due to resultant cerebral palsy, epilepsy, learning delays, and visual impairments [68]. In this condition, the brain is injured as a result of a lack of oxygen to the CNS due to complications during birth (such as cord prolapse, uterine rupture, breech presentation, and chorioamnionitis) [68]. However, inflammation is a major driver of the secondary energy failure that occurs about 6–120 h after the initial insult and is primarily driven by microglia and macrophages [42,69].

Interestingly, the NF-κB pathway becomes activated in hypoxic conditions, particularly in immunologic cell types like macrophages [43,70]. In line with this finding, hypoxia-induced NF-κB signaling has been recently identified in microglia within the CNS. Primary cultured microglia derived from a neonatal hypoxia-exposed rat model exhibited upregulation in phosphorylated NF-κB p65 proteins, proposed to be due to upregulation in Toll-Like Receptor 4 (TLR4, a significant inflammatory stimulus) expression [44]. Activation of microglial NF-κB in hypoxia has been linked to upstream signaling through both TLR-4 and Notch-1 signaling pathways [44,71]. Moreover, acute hypoxia was found to exacerbate NF-κB signaling in models of disease. Microglial NF-κB p65 expression was significantly upregulated in comparison to wildtype mice and mice under disease conditions in a transgenic rodent model simulating Alzheimer’s Disease [72]. As a physiological stressor, hypoxia is a critical stimulus of the NF-κB pathway in microglia, typically to detrimental effects.

Within in vivo models of HIE, specifically, NF-κB is a major mediator of microglia-induced inflammation. The selective knockout of NF-κB in microglia in a rodent model of neonatal hypoxic ischemic injury attenuated brain injury [45]. Phenotypically, this was evidenced through decreased ventriculomegaly and proinflammatory cytokine presentation in combination with improved locomotor outcomes as compared to controls [45]. The timing of NF-κB activation may also influence the developmental outcomes of HIE. NF-κB was chemically inhibited throughout the brain at various intervals in a neonatal rat model of hypoxic ischemic injury [73]. When the selective inhibitor of the NF-κB essential modulator (NEMO/IKKγ) was administered during the early phases (0–3 h post injury), microglial activation was completely prevented and the expression of proinflammatory cytokines (TNF, IL-1β) was attenuated. However, NF-κB inhibition during the later stages of injury (6-12 h post injury) was not sufficient to significantly reduce these indicators of inflammation. Fundamentally, this suggests that NF-κB-mediated signaling could be harnessed to reduce injury to the CNS in hypoxic ischemic encephalopathy.

## 4. Conclusions

The intricate relationship between the NF-κB pathway and microglial activation is gradually receiving more attention within the scope of research into neuroinflammation. The response of microglia to NF-κB-induced activation is particularly emphasized, especially in the context of cell and animal models representative of various neurodegenerative conditions. Microglia exhibit proinflammatory functional changes upon NF-κB activation, ultimately leading to the release of proinflammatory cytokines and chemokines which act on other cells in the central nervous system. Microglia-specific NF-κB inhibition is typically associated with positive downstream results, but more extensive research must be conducted into the intercellular and molecular interactions underlying this beneficial effect. Overall, the diverse effect of NF-κB on cell types within the central nervous system reviewed here supports the need for continued investigation into manipulating this molecular pathway for future neuroprotective therapies.

## Figures and Tables

**Figure 1 cells-13-00485-f001:**
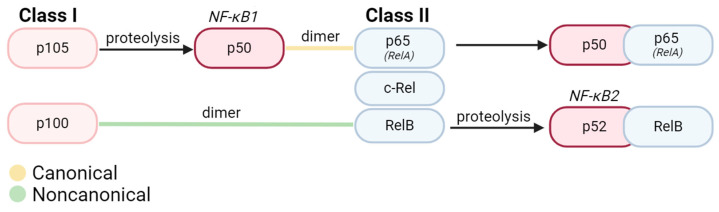
Structure of most abundant NF-κB dimers in canonical and noncanonical signaling. All proteins in the NF-κB family are comprised of 2 subunits. Broadly, Class I precursors must be proteolyzed into their active states and dimerized with another subunit (typically a Class II protein). In canonical signaling, Class I precursor p105 is proteolyzed into active NF-κB1 (p50) Upon proteolysis, these active subunits can bind to 3 known Class II proteins in the Rel family (p65/RelA, c-Rel, and RelB), bind to active p52, or homodimerize. Within noncanonical signaling, p100 first dimerizes with RelB before undergoing proteolysis for conversion into active NF-κB2 (p52).

**Figure 2 cells-13-00485-f002:**
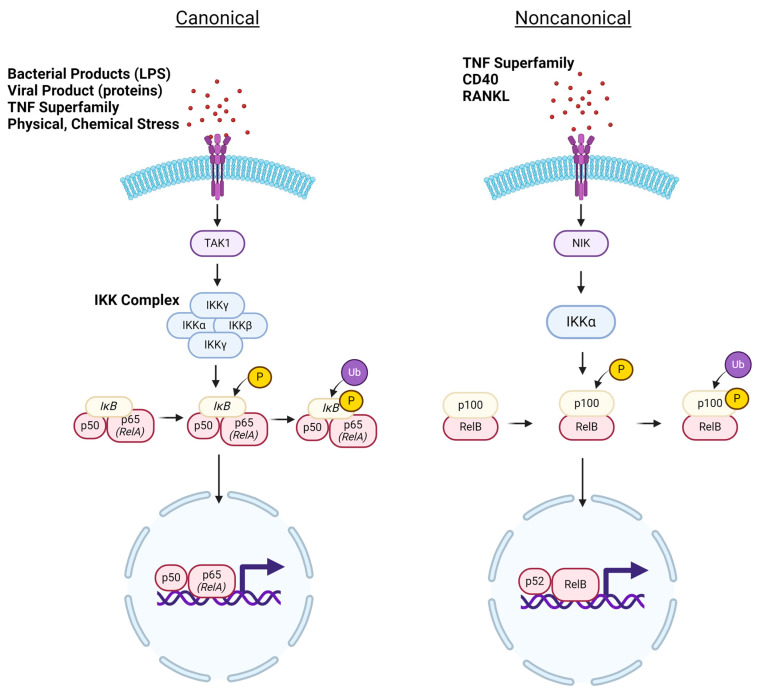
Canonical and noncanonical NF-κB signaling. Under homeostatic conditions, NF-κB exists in an inactivated state and is sequestered in the cytoplasm by IκB proteins. The canonical NF-κB pathway can be activated by a diverse range of immune-activating stimuli including bacterial or viral products, inflammatory cytokines, as well as physical or chemical stressors. These stimuli converge in TAK1 stimulation, which facilitates activation of the IKK complex, a tetramer composed of IKKα, IKKβ, and the regulatory IKKγ subunit. The activated IKK complex mediates phosphorylation of the IκB protein, enabling its eventual ubiquitination to allow for the release of active NF-κB. Stimuli inducing noncanonical signaling include proteins within the TNFR and RANKL families. In noncanonical signaling, the first major activated protein is NIK, which stimulates IKKα to phosphorylate the p100 subunit, leading to the eventual ubiquitination of the p100 subunit. This allows for the activated NF-kB transcription factor to translocate into the nucleus to influence gene expression.

**Table 1 cells-13-00485-t001:** Role of NF-κB activation in cell types found in the central nervous system.

Cell Type	NF-κB		
	Protective (against)	Detrimental (in)	Potential Roles
Neuron	Oxidative Stress [5,16]	Ischemia [23,24,25]	In survival [26,27]
			In development [27,28]
Oligodendrocyte	Inflammatory Cytokines [19,20]		In survival (possible) [29,30]
	Multiple Sclerosis (EAE) [21,31]		
Astrocyte		Spinal Cord Injury [32,33,34]	Differentiation [35]
Microglia		Alzheimer’s Disease [36,37,38]	
		Amyotrophic Lateral Sclerosis (SOD1 mutant) [39,40,41] Hypoxic Ischemic Encephalopathy [42,43,44,45]	Differentiation [46] Homeostatic Density [47]

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
