# Peer review of "NF-κB as an Inducible Regulator of Inflammation in the Central Nervous System"

_cells, 2024, doi:10.3390/cells13060485_

Round 1

Reviewer 1 Report

Comments and Suggestions for Authors

This review describes the role of the transcription factor NF-kB in CNS in health and disease. The review is succinct but clearly points out the sometimes-unexplainable opposing roles of this factor: beneficial in some instances whereas detrimental in other instances. 

A major concern in this review is the description of the activation pathway in a manner that is very confusing to a general reader (Fig. 1 and Fig. 2). In Fig. 1, if proteolysis is needed to release p50 and p52, then the precursor is a single protein. But the authors state later in Fig. 2 that p52 is kept inactive by the p100 subunit. Both cannot be true; it must be either the former or the latter. In Fig. 1, it is stated that for p50 and p52 to form dimers with class II proteins, they have to be generated from their precursors by proteolysis, meaning the the precursor proteins themselves are not capable of dimerization with class II proteins. But in Fig. 2, the authors depict dimerization of the precursor protein with RelB. This is contradictory.

In the non canonical pathway, what is the role of phosphorylation by activated IKKalpha? What happens when the p52 precursor protein gets phosphorylated? Does it promote proteolysis to release p52 for subsequent dimerization with RelB? It is depicted as if the phosphorylation promotes ubiqutination and possibly subsequent proteasomal degradation. If this were to be true, the entire precursor protein will be degraded. But Fig. 2 suggests that phosphorylation and ubiquitination somehow facilitates the generation of p52 for subsequent nuclear translocation.

How do activated TAK1 and NIK elicit their effects on their downstream targets IKK complex and IKKalpha? Is it by phosphorylation? In the canonical pathway, when IKK complex gets activated, it is understandable that it phosphorylates IkB for subsequent ubiquitination and proteasomal degradation. But the way the Fig. 2 is drawn, it is difficult to see that it is IkB that is phosphorylated and ubiquitinated. Someone might interpret the figure such that p50 is phosphorylated and p65 is ubiquitinated. It might be better if the figure is revised appropriately to avoid this potential confusion.

In Fig. 2 (non canonical pathway), is it THF-alpha or TNFR? It must be the former, and the authors need to correct this error. They also need to specify what the cell-surface receptor for DAMPs and PAMPs in this figure.

Is there any role for DAMPs and PAMPs in terms of their action with intracellular receptors? The way the authors describe the phenomenon, one might think that the only NF-kB is activated via DAMPs and PAMPs is via the cell-surface receptors. 

The authors need to carefully revise this background section and the two figures accordingly because without a clear description of the processes involved in the signaling pathway, the rest of the review becomes pointless and difficult to comprehend.

What are DAMPs? It is understood that these are secreted by damaged or dying cells. How can this include TNF-alpha, cytokines and chemokines? I thought that DAMPS when they activate cells via their receptors induce the secretion of TNF-alpha, cytokines and chemokines. But the authors state that TNF-alpha, cytokines and chemokines are themselves DAMPs. The authors need to check this and make appropriate corrections and provide relevant references for the revised section.

When describing the references 40 and 41, the authors state that the outcome of NF-kB in ischemia versus stroke is opposite to one another, but they talk about only the negative effect in both instances. Where are the opposing effects. 

Reviewer 2 Report

Comments and Suggestions for Authors

This is a review of the role that NFkB signaling plays in various immune pathways.  The review overall was well put together and written in an organized and comprehensive manner.  Figures were nicely laid out and appropriately referenced throughout the manuscript.  Some minor comments for the authors to address:

1. Line 29: should be homo- or heterodimers

2. Table 1: add references within the table for each of the phenotypes

3. Line 215: This is often dependent on the length of time that NFkB is active in the oligodendrocytes. For short activity it is protective however chronic activation is typically seen as detrimental.

Comments on the Quality of English Language

No issues with writing
